# Hyperspherical Prototype Node Clustering

**Jitao Lu**  *dianlujitao@gmail.com*
*School of Computer Science, School of Artificial Intelligence, OPtics and ElectroNics (iOPEN),*
*Northwestern Polytechnical University*

**Danyang Wu**  *danyangwu.cs@gmail.com*
*State Key Laboratory for Manufacturing Systems Engineering, School of Electronic and Information Engineering,*
*Xi'an Jiaotong University*

**Feiping Nie**[*]  *feipingnie@gmail.com*
*School of Artificial Intelligence, OPtics and ElectroNics (iOPEN), Northwestern Polytechnical University*

**Rong Wang**  *wangrong07@tsinghua.org.cn*
*School of Artificial Intelligence, OPtics and ElectroNics (iOPEN), Northwestern Polytechnical University*

**Xuelong Li**  *li@nwpu.edu.cn*
*School of Artificial Intelligence, OPtics and ElectroNics (iOPEN), Northwestern Polytechnical University*

**Reviewed on OpenReview:** *https://openreview.net/forum?id=z3ZlnaOMOd*

## Abstract

The general workflow of deep node clustering is to encode the nodes into node embeddings via graph neural networks and uncover clustering decisions from them, so clustering performance is heavily affected by the embeddings. However, existing works only consider preserving the semantics of the graph but ignore the inter-cluster separability of the nodes, so there's no guarantee that the embeddings can present a clear clustering structure. To remedy this deficiency, we propose **H**yperspherical **P**rototype **N**ode **C**lustering (HPNC), an end-to-end clustering paradigm that explicitly enhances the inter-cluster separability of learned node embeddings. Concretely, we constrain the embedding space to a unit-hypersphere, enabling us to scatter the cluster prototypes over the space with maximized pairwise distances. Then, we employ a graph autoencoder to map nodes onto the same hypersphere manifold. Consequently, cluster affinities can be directly retrieved from cosine similarities between node embeddings and prototypes. A clustering-oriented loss is imposed to sharpen the affinity distribution so that the learned node embeddings are encouraged to have small intra-cluster distances and large inter-cluster distances. Based on the proposed HPNC paradigm, we devise two schemes (HPNC-IM and HPNC-DEC) with distinct clustering backbones. Empirical results on popular benchmark datasets demonstrate the superiority of our method compared to other state-of-the-art clustering methods, and visualization results illustrate improved separability of the learned embeddings.

## 1 Introduction

Graph-structured data are ubiquitous in numerous domains, including social networks, recommendation systems, physics, and biology. Community structure is one of the most important characteristics of a graph and is useful in many downstream tasks. Finding communities from a graph can be formulated as a node clustering problem. Node clustering aims to categorize the nodes of a graph into a set of disjoint groups such that similar nodes are assigned to a common group. Spectral clustering (von Luxburg, 2007) has been one of the most successful and well-known node clustering methods in the past two decades. Given a

---

[*]Corresponding author.

weighted undirected graph, spectral clustering firstly uses Laplacian eigenmaps (Belkin & Niyogi, 2003) to embed the nodes into a low-dimensional feature space, then employs K-means to uncover clustering results from them. Being the first step of spectral clustering, the task of encoding nodes into feature vectors is called *node embeddings* (Cui et al., 2019), and Laplacian eigenmap is essentially the earliest node embedding method. Hence, the output of general-purpose node embedding methods can also be passed to K-means or other clustering methods for node clustering. However, real-world graph data are usually too complex for "shallow" embedding methods to capture the underlying relationships, leading to suboptimal performance of downstream tasks including clustering.

Recently, a surge in research on graph neural networks (GNNs) has led to state-of-the-art results on numerous downstream tasks. Moreover, the encoder can be easily chained with downstream tasks to customize the embeddings for them, which is non-trivial for previous methods. Methods employing a graph neural network (GNN) to do node embedding and perform clustering after that are called *deep clustering*. For example, graph autoencoder (GAE, Kipf & Welling 2016) uses stacked graph convolution network (GCN, Kipf & Welling 2017) layers to encode the input graph and node features into a low-dimensional space, then reconstructs the graph by inner products of the latent embeddings and minimizes the reconstruction error to self-supervise the training. After converging, the embeddings are passed to K-means to obtain clustering results. Later autoencoder-based approaches (Wang et al., 2017; Pan et al., 2020; Cui et al., 2020; Wang et al., 2020; Hou et al., 2022; Zhang et al., 2022) improved the GNN towards better latent embeddings. However, they mostly focus on having the learned embeddings better reconstruct the input graph and/or node features. That is, optimizing their objectives does not explicitly lead to better clusterability, so the resulting embeddings may not be suitable for clustering. To be specific, these methods overlooked the distribution of cluster prototypes[1]: *different cluster prototypes should be far from each other for their affiliated node embeddings to be distinguishable.* A concurrent work (Liu et al., 2023b) noticed this issue and proposed to push away different cluster centroids by penalizing the smallest centroid distance, but this is implemented as a regularization term so the effect is not guaranteed in practice.

On the other hand, most existing works either adopt the same sequential pipeline as GAE or follow (Yang et al., 2017) to iteratively optimize reconstruction loss and K-means loss. The fact is not limited to autoencoder-based approaches but also contrastive-learning-based approaches (Park et al., 2022; Devvrit et al., 2022; Liu et al., 2022a) that becoming popular in recent years. Exceptions are DAEGC (Wang et al., 2019) and SDCN (Bo et al., 2020). They pretrain the GAE as usual in the first step, then use K-means to obtain cluster centroids from the latent embeddings and follow (Xie et al., 2016) to form soft labels based on sample–centroid distances. After that, the embeddings are refined by self-training, and final clustering results are obtained from the soft labels after refinement. SDCN additionally normalized the latent embeddings by softmax function so soft labels can also be obtained there, but K-means is still unavoidable to obtain the centroids. The most critical drawback of K-means in a deep learning pipeline is that K-means' objective is not differentiable. As a result, it cannot be optimized by gradient descent thus prohibiting chained training with other downstream tasks. Moreover, the network parameters and clustering centroids have to be alternatively updated as the original K-means does, which leads to error propagation, prevents parallelism, and is prone to get stuck in bad local optimums.

In order for the learned embeddings to present a clear clustering structure, their intra-cluster distance should be as small as possible and inter-cluster distance should be as large as possible. Motivated by this criterion, this work mainly focuses on enhancing the separability of different clusters in unsupervised settings. In fact, this is similar to the objective of distance metric learning (Wang et al., 2014; Zhao et al., 2021b), which treats samples from the same class as positive pairs and samples from different classes as negative pairs and trains the neural network to generate similar embeddings for positive pairs and dissimilar embeddings for negative pairs. For instance, Proxy-NCA loss (Movshovitz-Attias et al., 2017) assigns a proxy to each class. During training, a data point is encouraged to be close to its corresponding class proxy, and far apart from other class proxies. As long as the proxies are separable, the learned embeddings are also separable. However, these class proxies are selected according to ground truth labels of the training set, so it's impossible to follow its selection strategy in unsupervised tasks and metric learning methods cannot be trivially adopted in clustering. Nonetheless, our method is heavily inspired by Proxy-NCA.

---

[1]Also known as *cluster centroids* in some clustering methods.

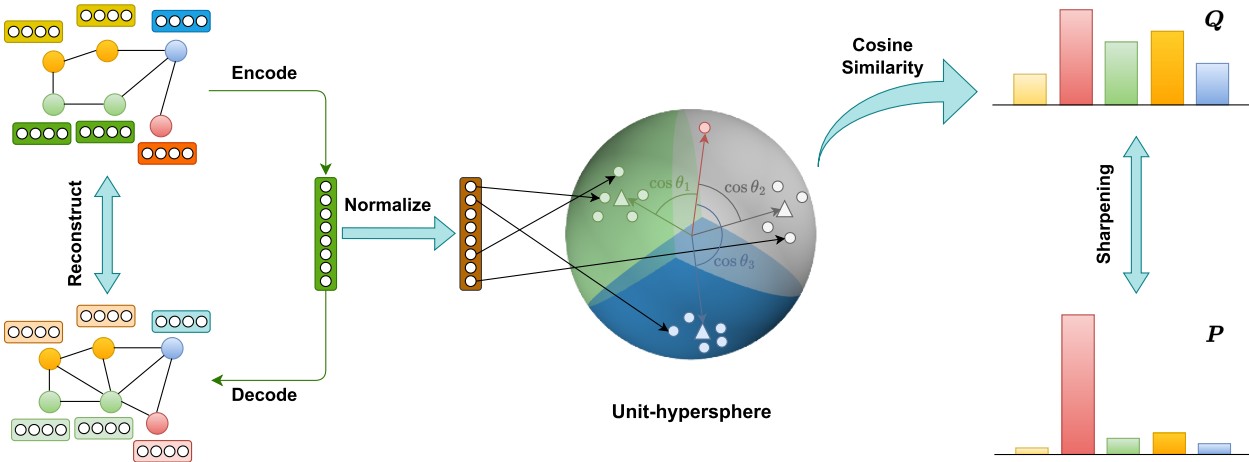

Figure 1: The workflow of HPNC. The input nodes are passed to several GNN layers to generate node embeddings. Then, the embeddings are passed to a GNN layer to reconstruct node features and an inner product decoder to reconstruct edges. $L^2$-normalization is applied to make them fit on a unit-hypersphere manifold. Given uniformly distributed cluster prototypes on the same manifold, the soft labels $Q$ can be obtained by the cosine similarities between node embeddings (gray circle) and cluster prototypes (colored triangles). A clustering loss is imposed on $Q$ to sharpen it so that confident predictions are emphasized and unconfident predictions are suppressed. The clustering loss is jointly minimized with the reconstruction errors towards discriminative node embeddings. Clustering results are directly obtained from $Q$.

In this paper, we simultaneously address the issues mentioned above and propose **H**yperspherical **P**rototype **N**ode **C**lustering (HPNC), a *fully differentiable* and *end-to-end* clustering paradigm which explicitly considers the separability of different clusters. Unlike previous works that infer clustering centroids from preliminary node embeddings, we use predefined centroids and maximize their margins to encourage the separability of different clusters. To this goal, we constrain the embedding space to a unit-hypersphere where the maximum distance between points is bounded. Then, we scatter the cluster prototypes on the hypersphere manifold so they have equal and maximized pairwise distances. After that, we use a graph autoencoder to map the input nodes onto the same manifold. Consequently, the soft labels can be obtained by their cosine similarities between prototypes. A clustering-oriented objective is jointly optimized with GAE to push nodes to corresponding prototypes. Remarkably, our HPNC is a general pipeline and can be used with any unsupervised representation learning methods and any deep clustering objective for joint representation learning and clustering. We demonstrate such flexibility by devising two clustering schemes based on HPNC. The general pipeline of HPNC is illustrated in Figure 1. Our main contributions are:

- We propose a novel node clustering paradigm called HPNC for joint representation learning and node clustering. HPNC explicitly maximizes the inter-cluster distances of the learned embeddings to provide sufficient discriminative power for clustering. Moreover, HPNC is fully differentiable so it can be easily integrated into a deep learning pipeline and jointly optimized with other modules.

- We develop a learning-based prototype rotation strategy to assist GNN to find matching prototypes. It allows the predefined prototypes to rotate on the hypersphere manifold so HPNC becomes less sensitive to their initial coordinates. Ablation study results verify that it's crucial to the clustering performance.

- Based on the proposed HPNC paradigm, we devise two schemes (*i.e.*, HPNC-IM and HPNC-DEC) to demonstrate the flexibility of HPNC to work with different clustering backbones. HPNC can also combine with other backbones to enjoy their advantages.

- Empirical results on widely adopted node clustering benchmark datasets consistently verify the effectiveness of our proposal compared to other state-of-the-art methods. To explore the characteristics of

the learned embeddings, we further apply K-means clustering and t-SNE visualization on them, and the results significantly reveal that our method indeed leads to more discriminative and separable latent embeddings.

## 2 Related Work

In this section, we briefly review the techniques closely related to our work, including unsupervised graph representation learning and deep clustering methods.

### 2.1 Unsupervised Graph Representation Learning

Unsupervised graph representation learning (UGRL) aims at projecting the nodes, edges, subgraphs, or entire graph, to a low-dimensional vector $\mathbb{R}^m$, without access to ground truth labels, so that they can be effectively handled by downstream machine learning algorithms. Traditional UGRL methods utilize matrix factorization (Belkin & Niyogi, 2003; Cao et al., 2015) and random walks (Grover & Leskovec, 2016) to capture graph characteristics. Specifically, DeepWalk Perozzi et al. (2014) defines the similarity between two nodes as the probability that they co-occur on a random walk, so it first estimates these probabilities from fixed-length and unbiased random walks sampled over the graph, then employs the skip-gram model to learn node embeddings that reconstruct these probabilities. node2vec Grover & Leskovec (2016) improved the walk strategy by using flexible, biased random walks that can trade off between local and global views of the graph. NetRA Yu et al. (2018b) proposed to learn the node embeddings with adversarially regularized autoendoers. Unlike skip-gram based models, it employs a long short-term memory (LSTM) network to map one hot vertex encodings into latent representations and train them to reconstruct the input. In addition to static graphs, NetWalk Yu et al. (2018a) proposed a novel reservoir sampling strategy to incrementally maintain the learned embeddings to effectively handle dynamic graphs. However, these methods utilize the graph topology only. For attributed graphs where each node is also associated with a feature vector, they are unable to exploit such extra information thus leading to suboptimal performance on downstream tasks. With the advance of deep learning on graphs, recent UGRL methods employ a GNN to perform the projection and design various unsupervised objectives to train the GNN. Autoencoder-based methods are the most popular thanks to their simplicity and effectiveness. GAE and VGAE (Kipf & Welling, 2016) take the dot products of latent embeddings as predicted edges and let them reconstruct the input graph. ARGA and ARVGA (Pan et al., 2020) introduced an adversarial regularizer on top of GAE and VGAE to encourage the latent embeddings to also follow a prior distribution. MGAE (Wang et al., 2017) passes corrupted node features to the GNN and lets the latent embeddings reconstruct uncorrupted node features. GALA (Park et al., 2019) designed a decoder to reconstruct node features from latent embeddings. Another fruitful avenue of UGRL is based on contrastive learning (Zhu et al., 2021), which learns meaningful embeddings by pulling predefined positive pairs and pushing negative pairs. DGI (Velickovic et al., 2019) generates graph embeddings of the input graph and its corrupted counterpart, then treats them as positive and negative samples of the uncorrupted node embeddings, respectively. InfoGraph (Sun et al., 2020) extends DGI to handle a batch of graphs, node embeddings from the same graph are positive pairs, and vice versa. MVGRL (Hassani & Ahmadi, 2020) employs graph diffusion (Klicpera et al., 2019) to generate an augmented view of the graph, then regards node embeddings from one view and graph embedding of another view as positive pairs. GRACE (Zhu et al., 2020) and SimGRACE (Xia et al., 2022) generate two augmented views of the graph, then follow the instance discrimination task (Wu et al., 2018) to treat a pair of embeddings of the same node in different views as positive, and all other pairs as negative. GGD (Zheng et al., 2022) simplified DGI and improved it to scale to large data. Compared to traditional UGRL methods, GNN-based methods usually incorporate the node attributes when learning latent embeddings, hence performing better on downstream tasks.

### 2.2 Deep Clustering

Real-world data are usually high-dimensional and non-globular, so shallow clustering methods such as K-means can not effectively handle them. Methods employing a deep neural network (DNN) to do non-linear mapping and perform clustering after that are called *deep clustering*, whose assumption is that mapped data distributions are more suitable for clustering than raw features. However, directly combining DNN

with K-means is not a viable approach, because the global optimum of K-means is achieved when DNN maps all data samples to one single point, which is called *cluster collapse.* To avoid this issue, most works employ an autoencoder to do the feature mapping, because the autoencoder's objective requires the latent embeddings to reconstruct the original inputs, which ensures that critical semantic information is preserved. GraphEncoder (Tian et al., 2014) is the earliest deep clustering method, it sequentially uses an autoencoder to perform feature mapping and K-means to cluster the latent embeddings. Beyond that, DCN (Yang et al., 2017) summed up K-means and autoencoder objectives and iteratively optimize their parameters, indirectly avoiding cluster collapse by minimizing autoencoder's reconstruction error. DEC (Xie et al., 2016) developed a self-supervised loss to iteratively refine autoencoder embeddings by confident predictions. Other than autoencoders, there are also deep clustering methods based on spectral clustering (Shaham et al., 2018), subspace clustering (Ji et al., 2017), mixture of experts (Tsai et al., 2021), *etc.* We suggest the readers refer to (Zhou et al., 2022) for a comprehensive survey on deep clustering methods.

As described above, the basic assumption of deep clustering is that the learned representations are more suitable for clustering, *i.e.*, more distinguishable from different clusters compared to raw features. Nonetheless, we argue that the assumption hardly holds for previous deep clustering methods. First, the representation learning modules aim at preserving the semantics (Kipf & Welling, 2016; Bo et al., 2020) or consistency (Bachman et al., 2019; Ji et al., 2019) of the latent embeddings, or making the embeddings discriminative at *instance* level (Zhao et al., 2021a; Tao et al., 2021; Devvrit et al., 2022). Second, clustering centroids and soft labels are inferred from the learned embeddings and (optionally) employed to finetune the embeddings. Unfortunately, none of the representation learning objectives contribute to *cluster-wise* discrimination due to inaccessible ground truth labels, and hence the ambiguous clustering centroids and soft labels, leading to suboptimal clustering performance. In a word, the chicken or the egg causality dilemma prohibits conventional deep clustering from learning unambiguous clustering results.

## 3 Method

An attributed undirected graph dataset $\mathcal{G} = \langle \mathcal{V}, \mathcal{E}, \boldsymbol{X} \rangle$ consists of a node set $\mathcal{V}$, an edge set $\mathcal{E}$, and a tabular data matrix $\boldsymbol{X} \in \mathbb{R}^{n \times d}$. Each node is associated with a feature vector $\boldsymbol{x}_i \in \mathbb{R}^d$. The edge set $\mathcal{E}$ can also be represented by an adjacency matrix $\boldsymbol{A}$, where

$$A_{u,v} = \begin{cases} 1, & \text{if } \langle u, v \rangle \in \mathcal{E}, \\ 0, & \text{otherwise.} \end{cases} \tag{1}$$

Node clustering algorithms aim to partition the nodes into $c$ disjoint clusters without ground truth labels so that nodes from the same cluster are similar to each other, and nodes from different clusters are not.

Next, we give an overview of the proposed HPNC method and analysis how it differs from conventional deep clustering diagrams. After that, we introduce its pipeline and details of each submodule.

### 3.1 Overview

Intuitively, cluster prototypes should be as far as possible from each other, so that the data samples distributed around them are more distinguishable from different clusters. Unlike previous works that infer cluster prototypes from the learned node embeddings in a K-means-like fashion (Xie et al., 2016; Devvrit et al., 2022), we directly maximize their distances in a *data-independent* approach. However, it's impossible to give the coordinates of the farthest pair of points in unconstrained Euclidean space, because the distances are up to infinity. Hence, the embedding space needs to be constrained to make the distances bounded. In this work, we simply restrict the embedding space to a unit hypersphere, which is widely adopted and massively studied in various machine learning tasks (Davidson et al., 2018; Xu et al., 2019; Wang & Isola, 2020; Tan et al., 2022). On a hypersphere manifold, the sum of pairwise distances of cluster prototypes is maximized when uniformly scattered, which implies the best separability among different clusters. Once obtaining the prototypes, their coordinates are fixed and no longer updated during training, so that their separability is always guaranteed and the cluster collapse issue is thoroughly avoided.

### 3.2 Hyperspherical Prototype Node Clustering

Next, we elaborate on the pipeline of HPNC. It merely consists of a representation learning module to map the nodes onto the same hypersphere manifold as the cluster prototypes, and a clustering module to push them close to corresponding prototypes. In this way, the intra-cluster distances are also naturally minimized, leading to discriminate embeddings. Moreover, the clustering labels can be directly retrieved from indexes of the nearest prototypes for each node, without reliance on external clustering algorithms.

#### 3.2.1 Pretrain Prototypes

The prerequisite step of HPNC is to obtain the coordinates of the cluster prototypes. Specifically, we propose to use uniformly scattered points on a unit-hypersphere as cluster prototypes. Uniformly placing $c$ points on a $m$-dimensional unit-hypersphere $\mathbb{S}^{m-1}$ so that the minimum distance between any two points gets maximized is known as the Tammes problem (Tammes, 1930) in geometry. For $\mathbb{S}^1$ (*i.e.*, a unit-circle on a 2D plane), this is as easy as splitting it into $c$ equal slices, and there are optimal solutions for $\mathbb{S}^2$ as well. Unfortunately, no such solutions exist for $m > 3$ given an arbitrary $c$ (Musin & Tarasov, 2015). To this end, we follow (Mettes et al., 2019) to adopt a gradient-based approach. Concretely, we minimize the maximum pairwise cosine similarities by gradient descent:

$$\mathcal{L}_{\text{HP}} = \frac{1}{c} \sum_{i=1}^{c} \max_{\boldsymbol{\mu}_j} \frac{\boldsymbol{\mu}_i^\top \boldsymbol{\mu}_j}{\|\boldsymbol{\mu}_i\| \cdot \|\boldsymbol{\mu}_j\|}, \; j \in 1, \ldots, c, j \neq i, \tag{2}$$

where $\boldsymbol{\mu}_i \in \mathbb{R}^m$ are unnormalized prototypes. For convenience, we use $\tilde{\boldsymbol{\mu}}_i = \frac{\boldsymbol{\mu}_i}{\|\boldsymbol{\mu}_i\|}$ to denote $L^2$-normalized prototypes. Thus, $\tilde{\boldsymbol{\mu}}_i$ will uniformly scatter on $\mathbb{S}^{m-1}$ after convergence. It's worth noting that the optimization is very fast because there are just $c \times m$ scalars to update. The pretraining is data-independent and thus only needs to be done once for each pair of $\langle c, m \rangle$. It's worthwhile to emphasize again that the prototypes are *fixed* after pretraining, they work as constants and are no longer updated in later steps of HPNC.

#### 3.2.2 Representation Learning

The next step is to map the input graph into low-dimensional node embeddings. Without access to ground truth labels in a clustering context, an unsupervised objective is demanded to produce meaningful embeddings. Motivated by recent advances in generative graph representation learning, we adopt a masked graph autoencoder (GraphMAE) (Hou et al., 2022) as the representation learning backbone. We briefly introduce its objective in this section.

**Masked feature reconstruction.** Unlike conventional GAEs that reconstruct edges, GraphMAE developed a masked feature reconstruction strategy to recover masked node features $\boldsymbol{X} \in \mathbb{R}^{n \times d}$. To be specific, a large random subset (*e.g.*, 50%) of nodes $\mathcal{V}_{\text{mask}} \in \mathcal{V}$ are sampled, and their node features are replaced by a shared, trainable *mask token*. Then, the partially masked features and unaltered input graph are fed into multiple GNN layers to generate latent embeddings $\boldsymbol{Z} \in \mathbb{R}^{n \times m}$. A re-mask strategy is applied to $\boldsymbol{Z}$ before feeding it to the decoder, which replaces the embeddings of $\mathcal{V}_{\text{mask}}$ again, but with zeros. Denoting the reconstructed node features as $\hat{\boldsymbol{X}} \in \mathbb{R}^{n \times d}$, GraphMAE introduced scaled cosine error to minimize the reconstruction error as

$$\mathcal{L}_{\text{fea}} = \frac{1}{|\mathcal{V}_{\text{mask}}|} \sum_{v_i \in \mathcal{V}_{\text{mask}}} \left( 1 - \frac{\boldsymbol{x}_i^\top \hat{\boldsymbol{x}}_i}{\|\boldsymbol{x}_i\| \cdot \|\hat{\boldsymbol{x}}_i\|} \right)^\gamma, \gamma \geqslant 1. \tag{3}$$

During inference, the original node features without mask and re-mask strategies are used. As a result, it potentially leads to a mismatch between training and interference because the mask token is not observed in the later stage. To mitigate this issue, GraphMAE adopts a "random-substitution" method that randomly substitutes node features of a small subset of $\mathcal{V}_{\text{mask}}$ (*e.g.*, 15%) with node features sampled from $\boldsymbol{X}$.

**Edge reconstruction.** GraphMAE focuses on classification and the authors argue that feature reconstruction is more beneficial than edge reconstruction (Hou et al., 2022), so they propose to reconstruct node features only. However, reconstructing the edges $\boldsymbol{A}$ captures more node pair-level information (Liu et al.,

2023a) and thus beneficial for link prediction and clustering (Pan et al., 2020). Hence, we also reconstruct the edges in addition to node features (Gao & Huang, 2018; Salehi & Davulcu, 2020). To be specific, we adopt the widely used inner product decoder (Kipf & Welling, 2016) to predict edges based on the latent embeddings:

$$A_{u,v} = \sigma \left( \frac{\boldsymbol{z}_u^\top \boldsymbol{z}_v}{\|\boldsymbol{z}_u\| \cdot \|\boldsymbol{z}_v\|} \right), \tag{4}$$

where $\sigma(\cdot)$ is the sigmoid function. Then, we measure their discrepancies between ground truth edges by binary cross entropy as

$$\mathcal{L}_{\text{edge}} = -\frac{1}{|\mathcal{E}|} \left( \sum_{\langle u,v \rangle \in \mathcal{E}} \log A_{u,v} + \sum_{\langle \bar{u},\bar{v} \rangle \in \bar{\mathcal{E}}} \log(1 - A_{\bar{u},\bar{v}}) \right), \tag{5}$$

where $\bar{\mathcal{E}}$ is the set of unconnected edges obtained by negative sampling and $|\mathcal{E}| = |\bar{\mathcal{E}}|$. By applying just $\mathcal{L}_{\text{fea}}$ and $\mathcal{L}_{\text{edge}}$, the GNN can generate semantically meaningful node embeddings, and clustering results can be obtained by applying external clustering algorithms such as K-means. Next, we describe how to further refine them for better clustering performance and obtain clustering labels without reliance on K-means.

### 3.2.3 Rotated Clustering Affinity

In order to prevent collapsed clusters and encourage large inter-cluster distance of the latent embeddings, we aim to map nodes around scattered cluster prototypes while leaving these prototypes unchanged. However, we empirically find that it's too challenging for the GNN to map nodes to appropriate prototypes. To mitigate this issue, we propose allowing the prototypes to *rotate* on the hypersphere. With a shared rotation matrix, their pairwise similarities still remain unchanged so the large separation property is preserved, while their coordinates can change to some extent. In other words, the GNN doesn't need to learn to rotate the nodes anymore so it will converge easier. To be specific, we first apply $L^2$-normalization to latent embeddings and denote them as $\tilde{\boldsymbol{z}}_i = \frac{\boldsymbol{z}_i}{\|\boldsymbol{z}_i\|}$, then the cosine similarity of the $i$-th node and $j$-th rotated prototype is simply their inner product $\tilde{\boldsymbol{z}}_i^\top \boldsymbol{R} \tilde{\boldsymbol{\mu}}_j$, where $\boldsymbol{R} \in \mathbb{R}^{m \times m}, \boldsymbol{R}^\top \boldsymbol{R} = \boldsymbol{I}$ is the rotation matrix. Finally, we apply the softmax function to make them sum up to 1:

$$Q_{i,j} = \frac{\exp(\tilde{\boldsymbol{z}}_i^\top \boldsymbol{R} \tilde{\boldsymbol{\mu}}_j)}{\sum_{j'=1}^{c} \exp(\tilde{\boldsymbol{z}}_i^\top \boldsymbol{R} \tilde{\boldsymbol{\mu}}_{j'})}, \quad \text{s.t. } \boldsymbol{R}^\top \boldsymbol{R} = \boldsymbol{I}. \tag{6}$$

Hence, $\boldsymbol{Q} \in \mathbb{R}^{n \times c}$ can be interpreted as (soft) clustering labels. The rotation matrix $\boldsymbol{R}$ is learnable and updated by gradient descent[2]. After obtaining preliminary clustering affinities, we devise two schemes to refine them (hence the node embeddings) as we elaborate below.

### 3.3 Scheme 1: HPNC-IM

The clustering affinities $\boldsymbol{Q}$ should be unambiguous and sharp for $\tilde{\boldsymbol{Z}}$ to be discriminative, so that intra-cluster distances are minimized and inter-cluster distances are maximized. To this end, we can employ the *information maximization (IM)* loss (Gomes et al., 2010; Hu et al., 2017; Liang et al., 2020) to encourage the cluster distributions individually unambiguous and globally uniform. Let $\mathcal{X}$ and $\mathcal{Y}$ denote the domains of data samples and cluster assignments, IM minimizes the following objective:

$$\mathcal{L}_{\text{IM}} = -(H(Y) - H(Y|X)), \tag{7}$$

---

[2]We initialize $\boldsymbol{R}$ as an identity matrix and enforce the orthogonal constraint with a PyTorch built-in function `torch.nn.utils.parametrizations.orthogonal` during training.

where $X \in \mathcal{X}$ and $Y \in \mathcal{Y}$ denotes the random variables for data samples and cluster assignments, $H(\cdot)$ and $H(\cdot|\cdot)$ are the marginal and conditional entropy, which can be estimated as

$$\mathcal{L}_{\text{bal}} = H(Y) = h(\boldsymbol{p}_\theta(\boldsymbol{y})) = h(\mathbb{E}_{\boldsymbol{x}_i \in \mathcal{X}}(\boldsymbol{p}_\theta(\boldsymbol{y}|\boldsymbol{x}_i))) = h\left(\frac{1}{n}\sum_{i=1}^{n} \boldsymbol{Q}_{i,:}\right) = \log c - D_{\text{KL}}\left(\mathbb{E}_{\boldsymbol{x}_i \in \boldsymbol{X}}(\boldsymbol{p}_\theta(\boldsymbol{y}|\boldsymbol{x}_i))\|\frac{1}{c}\boldsymbol{1}\right),$$
(8)

$$\mathcal{L}_{\text{ent}} = H(Y|X) = \mathbb{E}_{\boldsymbol{x}_i \in \mathcal{X}}(h(\boldsymbol{p}_\theta(\boldsymbol{y}|\boldsymbol{x}_i))) = \frac{1}{n}\sum_{i=1}^{n} h(\boldsymbol{Q}_{i,:}),$$
(9)

where $\boldsymbol{p}_\theta(\boldsymbol{y})$ and $\boldsymbol{p}_\theta(\boldsymbol{y}|\cdot)$ are the marginal and conditional probabilities over label values $\boldsymbol{y} \in \{1, \dots, c\}$ modeled by a GNN with parameters $\theta$, $h(\boldsymbol{p}) = -\sum_{j=1}^{c} p_j \log p_j$ denotes the entropy function, and $\boldsymbol{1}$ is a vector with all ones.

Intuitively, maximizing Eq. (8) encourages the average cluster probabilities $\mathbb{E}_{\boldsymbol{x}_i \in \mathcal{X}}(\boldsymbol{p}_\theta(\boldsymbol{y}|\boldsymbol{x}_i))$ to converge to the uniform distribution, so data samples tend to be evenly assigned to different clusters. On the other hand, minimizing Eq. (9) encourages individual cluster probabilities $\boldsymbol{p}_\theta(\boldsymbol{y}|\boldsymbol{x}_i)$ to approximate the one-hot distribution, so the cluster assignment becomes unambiguous, and the samples are pushed far from the decision boundaries. In summary, minimizing Eq. (7) trains the GNN to learn embeddings that agree with the *cluster assumption* (Chapelle & Zien, 2005), hence improving clustering performance. Combining Eqs. (3), (5) and (7) together, the full objective of HPNC-IM is

$$\mathcal{L}_{\text{HPNC-IM}} = \mathcal{L}_{\text{fea}} + \alpha\mathcal{L}_{\text{edge}} - \beta\mathcal{L}_{\text{bal}} + \gamma\mathcal{L}_{\text{ent}}.$$
(10)

Unlike previous works that require bootstrapping the autoencoder for hundreds of epochs with the reconstruction loss only, we randomly initialize the network parameters and train HPNC-IM with the full objective from scratch.

Finally, clustering results are directly obtained as the indices of the largest $Q_{i,j}$ without relying on external clustering algorithms such as K-means. Nonetheless, we'll show later in Section 4.2.2 that the clustering performance of $\boldsymbol{Q}$ and K-means on $\tilde{\boldsymbol{Z}}$ are very close, verifying that our objective actually leads to discriminative node embeddings.

### 3.4 Scheme 2: HPNC-DEC

In addition to the IM loss, we demonstrate that our proposed HPNC paradigm can also integrate with a simplified version of the DEC loss (Xie et al., 2016). As introduced in Section 1, the original version of DEC is a three-stage process:

1. Employing an autoencoder to learn preliminary embeddings $\{\boldsymbol{z}_1, \dots, \boldsymbol{z}_n\}$ from the input data.

2. Performing K-means on the embeddings to obtain clustering centroids $\{\boldsymbol{\mu}_1, \dots, \boldsymbol{\mu}_c\}$, then calculating the cluster assignment distribution $\boldsymbol{Q}$ with the Student's $t$-distribution as the similarity measurement between the embeddings and clustering centroids:

$$Q_{i,j} = \frac{(1 + \|\boldsymbol{z}_i - \boldsymbol{\mu}_j\|^2/\sigma)^{-\frac{\sigma+1}{2}}}{\sum_{j'}(1 + \|\boldsymbol{z}_i - \boldsymbol{\mu}_{j'}\|^2/\sigma)^{-\frac{\sigma+1}{2}}},$$
(11)

where $\sigma$ is the degree of freedom of the Student's $t$-distribution.

3. Defining an auxiliary distribution $\boldsymbol{P}$ by raising $\boldsymbol{Q}$ to the second power and normalizing it to sum up to 1:

$$P_{i,j} = \frac{Q_{i,j}^2/\sum_{i'=1}^{n} Q_{i',j}}{\sum_{j'=1}^{c}(Q_{i,j'}^2/\sum_{i'=1}^{n} Q_{i',j'})}.$$
(12)

Then, their Kullback-Leibler (KL) divergence is minimized to finetune the encoder network:

$$\mathcal{L}_{\text{DEC}} = D_{\text{KL}}(\boldsymbol{P}\|\boldsymbol{Q}) = \frac{1}{n}\sum_{i=1}^{n}\sum_{j=1}^{c} P_{i,j} \log \frac{P_{i,j}}{Q_{i,j}}.$$
(13)

$\boldsymbol{P}$ is sharper than $\boldsymbol{Q}$ because confident predictions (ones with large $Q_{i,j}$) are emphasized and unconfident predictions (ones with small $Q_{i,j}$) are suppressed. Hence, minimizing the KL divergence will sharpen $\boldsymbol{Q}$, leading to discriminative and cluster-aware embeddings. DEC iteratively performs 2) and 3) until convergence. However, the reliance on K-means and iterative optimization prohibits the potential use of DEC in an end-to-end deep learning pipeline.

We propose a crucial simplification to integrate DEC with the HPNC paradigm, that is to replace the Student's $t$-distribution-based cluster assignment distribution (*i.e.*, step 2) with Eq. (6). Benefiting from the pipeline of HPNC, we no longer need K-means because the clustering centroids are given in advance. The iterative optimization of centroids and encoder parameters is also discarded because the centroids are *not* updated during training HPNC.

Intuitively, DEC shares a similar purpose with the conditional entropy minimization part of IM (9) to make the target cluster assignment certain. However, it doesn't consider balancing the scale of different clusters so it may produce empty clusters, which is undesired when we would like to ensure that all clusters are assigned data samples. Thus, we simply employ the marginal entropy maximization part of IM (8) to prevent degenerate solutions with empty clusters. Finally, the full objective of HPNC-DEC is as follows:

$$\mathcal{L}_{\text{HPNC-DEC}} = \mathcal{L}_{\text{fea}} + \alpha\mathcal{L}_{\text{edge}} - \beta\mathcal{L}_{\text{bal}} + \gamma\mathcal{L}_{\text{DEC}}. \tag{14}$$

As HPNC-IM, HPNC-DEC is also trained from randomly initialized network parameters by end-to-end gradient descent, without any form of pretraining. The loss functions of both schemes Eqs. (10) and (14) are fully differentiable and thus can be easily integrated into a deep learning pipeline and jointly optimized with other modules.

### 3.5 Complexity Analysis

When pretraining prototypes, the time complexity of calculating their pairwise similarities is $\mathcal{O}(c^2m)$, and there are $cm$ parameters to be updated, so the complexity of pretraining is $\mathcal{O}(c^2mt_{pre})$ where $t_{pre}$ refers to the number of pretraining epochs. We empirically find that setting $t_{pre}$ to 3000 is sufficient, so this step usually finishes in half of a minite.

The complexity of representation learning depends on the specific backbone, which is usually linear with the number of nodes and edges.

The complexity of calculating rotated clustering affinity (6) is $\mathcal{O}(ncm + m^2c)$ by applying rotation to prototypes first and obtain their inner products with samples later. There are $m^2$ trainable parameters in $\boldsymbol{R}$, and its orthogonal parametrization is also $\mathcal{O}(m^2)$. Hence, this step needs $\mathcal{O}((n+m)cmt)$ in total, where $t$ is the number of training epochs.

The complexity of the clustering backbone also depends on the specific choice, which is $\mathcal{O}(nct)$ for both IM and DEC that employed in this work.

In summary, the time complexity of HPNC without representation learning is $\mathcal{O}(c^2mt_{pre} + (n+m)cmt)$. The time complexity of K-means clustering is $\mathcal{O}(ncmt)$, so HPNC is comparable to applying K-means after representation learning.

## 4 Experiments

In this section, we conduct extensive experiments to evaluate the effectiveness of the proposed HPNC paradigm. The experiments are designed with the aim to answer the following research questions:

- **RQ1:** What's the clustering performance of the proposed HPNC paradigm along with the two devised schemes?

- **RQ2:** How useful is the proposed *rotated clustering affinity* compared to conventional cosine similarity?

Table 1: Averaged clustering performance. The best results are in **bold**, second-best results are underlined.

| Method | Cora | | | CiteSeer | | | PubMed | | | ACM | | | DBLP | | |
|---|---|---|---|---|---|---|---|---|---|---|---|---|---|---|---|
| | ACC | NMI | ARI | ACC | NMI | ARI | ACC | NMI | ARI | ACC | NMI | ARI | ACC | NMI | ARI |
| GAE | $53.3_{\pm0.2}$ | $40.7_{\pm0.3}$ | $30.5_{\pm0.2}$ | $41.3_{\pm0.4}$ | $18.3_{\pm0.3}$ | $19.1_{\pm0.3}$ | $63.1_{\pm0.4}$ | $24.9_{\pm0.3}$ | $21.7_{\pm0.2}$ | $84.5_{\pm1.4}$ | $55.4_{\pm1.9}$ | $59.5_{\pm3.1}$ | $61.2_{\pm1.2}$ | $30.8_{\pm0.9}$ | $22.0_{\pm1.4}$ |
| VGAE | $56.0_{\pm0.3}$ | $38.5_{\pm0.4}$ | $34.7_{\pm0.3}$ | $44.4_{\pm0.2}$ | $22.7_{\pm0.3}$ | $20.6_{\pm0.3}$ | $65.5_{\pm0.2}$ | $25.0_{\pm0.4}$ | $20.3_{\pm0.2}$ | $84.1_{\pm0.2}$ | $53.2_{\pm0.5}$ | $57.7_{\pm0.7}$ | $58.6_{\pm0.1}$ | $26.9_{\pm0.1}$ | $17.9_{\pm0.1}$ |
| MGAE | $63.4_{\pm0.5}$ | $45.6_{\pm0.3}$ | $43.6_{\pm0.4}$ | $63.5_{\pm0.4}$ | $39.7_{\pm0.4}$ | $42.5_{\pm0.5}$ | $59.3_{\pm0.5}$ | $28.2_{\pm0.2}$ | $24.8_{\pm0.4}$ | $87.6_{\pm0.0}$ | $62.5_{\pm0.0}$ | $67.1_{\pm0.0}$ | $75.6_{\pm0.1}$ | $43.3_{\pm0.2}$ | $47.6_{\pm0.2}$ |
| ARGA | $63.9_{\pm0.4}$ | $45.1_{\pm0.3}$ | $35.1_{\pm0.5}$ | $57.3_{\pm0.5}$ | $35.2_{\pm0.3}$ | $34.0_{\pm0.4}$ | $68.0_{\pm0.5}$ | $27.6_{\pm0.4}$ | $29.0_{\pm0.4}$ | $86.3_{\pm0.4}$ | $56.2_{\pm0.8}$ | $63.4_{\pm0.9}$ | $64.8_{\pm0.6}$ | $29.4_{\pm0.9}$ | $28.0_{\pm0.9}$ |
| ARVGA | $64.0_{\pm0.5}$ | $44.9_{\pm0.4}$ | $37.4_{\pm0.5}$ | $54.4_{\pm0.5}$ | $26.1_{\pm0.5}$ | $24.5_{\pm0.3}$ | $51.3_{\pm0.4}$ | $11.7_{\pm0.3}$ | $7.8_{\pm0.2}$ | $83.9_{\pm0.5}$ | $51.9_{\pm1.0}$ | $57.8_{\pm1.2}$ | $54.4_{\pm0.4}$ | $25.9_{\pm0.3}$ | $19.8_{\pm0.4}$ |
| NetRA | $30.4_{\pm0.7}$ | $9.5_{\pm2.6}$ | $4.7_{\pm0.7}$ | $25.7_{\pm3.3}$ | $1.9_{\pm1.0}$ | $1.5_{\pm1.0}$ | $40.0_{\pm0.0}$ | $0.0_{\pm0.0}$ | $0.0_{\pm0.0}$ | $36.7_{\pm0.2}$ | $0.6_{\pm0.1}$ | $0.5_{\pm0.1}$ | $33.3_{\pm0.4}$ | $1.5_{\pm0.2}$ | $0.8_{\pm0.2}$ |
| NetWalk | $27.2_{\pm3.1}$ | $4.5_{\pm0.5}$ | $-0.2_{\pm0.2}$ | $24.9_{\pm0.2}$ | $3.5_{\pm0.2}$ | $1.9_{\pm0.1}$ | $35.8_{\pm2.3}$ | $0.18_{\pm0.3}$ | $0.2_{\pm0.3}$ | $37.9_{\pm0.4}$ | $1.6_{\pm0.4}$ | $0.9_{\pm0.0}$ | $34.0_{\pm0.5}$ | $2.0_{\pm0.3}$ | $1.8_{\pm0.0}$ |
| AGC | $68.9_{\pm0.5}$ | $53.7_{\pm0.3}$ | $48.6_{\pm0.3}$ | $66.9_{\pm0.5}$ | $41.1_{\pm0.4}$ | $41.9_{\pm0.5}$ | $69.8_{\pm0.4}$ | $31.6_{\pm0.3}$ | $31.8_{\pm0.4}$ | $79.9_{\pm0.0}$ | $49.6_{\pm0.0}$ | $51.2_{\pm0.0}$ | $64.4_{\pm0.1}$ | $34.6_{\pm0.0}$ | $28.2_{\pm0.1}$ |
| DAEGC | $70.2_{\pm0.4}$ | $52.6_{\pm0.3}$ | $49.7_{\pm0.4}$ | $67.2_{\pm0.5}$ | $39.7_{\pm0.5}$ | $41.1_{\pm0.4}$ | $66.8_{\pm0.5}$ | $26.6_{\pm0.2}$ | $27.7_{\pm0.3}$ | $86.9_{\pm2.8}$ | $56.2_{\pm4.2}$ | $59.4_{\pm3.9}$ | $62.1_{\pm0.5}$ | $32.5_{\pm0.5}$ | $21.0_{\pm0.5}$ |
| AGE | $72.8_{\pm0.5}$ | $58.1_{\pm0.6}$ | $\mathbf{56.3_{\pm0.4}}$ | $70.0_{\pm0.3}$ | $44.6_{\pm0.4}$ | $45.4_{\pm0.5}$ | $69.9_{\pm0.5}$ | $30.1_{\pm0.4}$ | $31.4_{\pm0.6}$ | $90.6_{\pm0.2}$ | $68.7_{\pm0.5}$ | $74.3_{\pm0.4}$ | $75.1_{\pm0.6}$ | $45.5_{\pm0.3}$ | $47.6_{\pm0.8}$ |
| SDCN | $48.5_{\pm0.5}$ | $24.6_{\pm0.4}$ | $20.6_{\pm0.3}$ | $66.0_{\pm0.3}$ | $38.7_{\pm0.3}$ | $40.2_{\pm0.4}$ | $64.2_{\pm1.3}$ | $22.9_{\pm2.0}$ | $22.3_{\pm2.0}$ | $90.5_{\pm0.2}$ | $68.3_{\pm0.3}$ | $73.9_{\pm0.4}$ | $68.1_{\pm1.8}$ | $39.5_{\pm1.3}$ | $39.2_{\pm2.0}$ |
| DCRN | $63.4_{\pm0.5}$ | $46.2_{\pm0.4}$ | $36.3_{\pm0.9}$ | $70.9_{\pm0.2}$ | $45.9_{\pm0.4}$ | $47.6_{\pm0.3}$ | $69.9_{\pm0.1}$ | $32.2_{\pm0.1}$ | $31.4_{\pm0.1}$ | $91.9_{\pm0.2}$ | $71.6_{\pm0.6}$ | $77.6_{\pm0.5}$ | $79.7_{\pm0.3}$ | $49.0_{\pm0.4}$ | $53.6_{\pm0.5}$ |
| GraphMAE | $68.0_{\pm2.0}$ | $57.6_{\pm1.1}$ | $50.2_{\pm2.8}$ | $69.0_{\pm0.4}$ | $43.6_{\pm0.5}$ | $44.4_{\pm0.5}$ | $69.9_{\pm0.5}$ | $\underline{34.4_{\pm0.5}}$ | $32.9_{\pm0.7}$ | $89.6_{\pm0.2}$ | $66.7_{\pm0.3}$ | $71.8_{\pm0.4}$ | $73.6_{\pm0.5}$ | $45.7_{\pm0.3}$ | $43.6_{\pm0.7}$ |
| SUBLIME | $71.2_{\pm0.1}$ | $53.6_{\pm0.4}$ | $50.6_{\pm0.4}$ | $68.1_{\pm0.5}$ | $43.2_{\pm0.3}$ | $43.4_{\pm0.7}$ | $63.8_{\pm1.3}$ | $27.4_{\pm1.4}$ | $24.5_{\pm2.3}$ | $88.9_{\pm0.3}$ | $66.0_{\pm0.8}$ | $70.1_{\pm0.7}$ | $54.8_{\pm2.9}$ | $30.2_{\pm2.6}$ | $18.1_{\pm2.4}$ |
| NAFS | $70.4_{\pm0.0}$ | $56.6_{\pm0.0}$ | $48.0_{\pm0.0}$ | $71.8_{\pm0.0}$ | $45.1_{\pm0.0}$ | $47.6_{\pm0.0}$ | $\underline{70.5_{\pm0.0}}$ | $33.9_{\pm0.0}$ | $33.2_{\pm0.0}$ | $81.2_{\pm0.1}$ | $51.4_{\pm0.3}$ | $52.9_{\pm0.2}$ | $52.8_{\pm0.0}$ | $25.7_{\pm0.1}$ | $14.7_{\pm0.0}$ |
| CONVERT | $74.1_{\pm1.5}$ | $55.6_{\pm1.1}$ | $50.5_{\pm2.0}$ | $68.4_{\pm0.7}$ | $41.6_{\pm0.7}$ | $42.8_{\pm1.6}$ | $67.1_{\pm1.8}$ | $30.0_{\pm1.5}$ | $29.3_{\pm1.8}$ | $84.4_{\pm2.9}$ | $55.3_{\pm4.4}$ | $59.6_{\pm6.8}$ | $52.9_{\pm2.8}$ | $20.4_{\pm2.1}$ | $17.2_{\pm2.36}$ |
| HPNC-IM | $\underline{74.1_{\pm0.5}}$ | $58.9_{\pm0.6}$ | $51.5_{\pm0.9}$ | $72.4_{\pm0.5}$ | $46.3_{\pm0.7}$ | $48.4_{\pm0.7}$ | $70.4_{\pm0.4}$ | $\mathbf{34.6_{\pm0.7}}$ | $\mathbf{33.5_{\pm0.7}}$ | $92.1_{\pm0.1}$ | $\underline{72.1_{\pm0.3}}$ | $78.1_{\pm0.3}$ | $80.0_{\pm0.4}$ | $\underline{49.8_{\pm0.4}}$ | $54.9_{\pm0.6}$ |
| HPNC-DEC | $\mathbf{74.4_{\pm0.5}}$ | $\mathbf{59.4_{\pm0.9}}$ | $\underline{51.9_{\pm1.2}}$ | $\mathbf{73.0_{\pm0.5}}$ | $\mathbf{46.9_{\pm0.7}}$ | $\mathbf{49.7_{\pm0.7}}$ | $\mathbf{70.6_{\pm0.9}}$ | $33.1_{\pm1.5}$ | $\underline{33.3_{\pm1.4}}$ | $\mathbf{92.3_{\pm0.1}}$ | $\mathbf{72.4_{\pm0.3}}$ | $\mathbf{78.4_{\pm0.2}}$ | $\mathbf{80.1_{\pm0.5}}$ | $\mathbf{49.9_{\pm0.5}}$ | $\mathbf{55.0_{\pm0.7}}$ |

- **RQ3:** Despite clustering assignments can be directly retrieved from the soft labels $\boldsymbol{Q}$, what's the difference between the ones obtained by performing K-means on the embeddings $\hat{\boldsymbol{Z}}$?

- **RQ4:** What's the relationship between the centroids found by performing K-means on $\hat{\boldsymbol{Z}}$ and the rotated HPNC prototypes $\{\boldsymbol{R}\tilde{\boldsymbol{\mu}}_j\}_{j=1}^c$?

- **RQ5:** Is the separability of the learned embeddings really improved?

**Baselines.**    We compare HPNC with 13 classical and state-of-the-art node clustering and node embedding models, including GAE, VGAE (Kipf & Welling, 2016), MGAE (Wang et al., 2017), ARGA, ARVGA (Pan et al., 2020), NetRA Yu et al. (2018b), NetWalk Yu et al. (2018a), AGC (Zhang et al., 2019), DAEGC (Wang et al., 2019), AGE (Cui et al., 2020), SDCN (Bo et al., 2020), DCRN (Liu et al., 2022b), GraphMAE (Hou et al., 2022), SUBLIME (Liu et al., 2022a), NAFS (Zhang et al., 2022), and CONVERT Yang et al. (2023). NetRA and NetWalk are random-walk-based node embedding methods that utilize graph topology only, and the rest competitors are designed to handle attributed graphs.

**Evaluation protocol.**    For node embedding methods that cannot directly produce cluster assignments, we perform K-means on the learned embeddings to obtain labels for comparison. K-means is reinitialized for 50 times to eliminate randomness, so that their clustering performance solely depends on the quality of the learned embeddings. We run all competitors five times with distinct random seeds and report the averaged results and standard deviations of the best epochs.

We present the selected datasets, evaluation metrics and implementation details in Appendices A.1 to A.3, respectively.

## 4.1   Node Clustering Performance Comparison (RQ1)

Table 1 reports the clustering performance of our method and other baselines. As shown in the table, both schemes of our proposed HPNC consistently achieved the best or second-best clustering performance on all five datasets, revealing the effectiveness of HPNC. In contrast, some competitors achieved appealing performance on certain datasets but failed on others. For instance, the ACC of AGE on Cora is only 1.6

lower than that of HPNC-DEC, but it failed to handle DBLP, and the ACC is 5.0 lower, which is a severe performance drop. DCRN's ACC on ACM is only 0.2 lower than that of HPNC-IM, but 10.7 lower on Cora. Generally speaking, the performance of HPNC is quite competitive compared with recent state-of-the-art models. Apart from clustering performance, HPNC also enjoys the benefits from its end-to-end training strategy: Other strong baselines including but not limited to AGE, GraphMAE, and NAFS all rely on K-means or spectral clustering to obtain clustering labels, increasing hardware requirements on CPU/memory and prohibiting end-to-end training with other downstream tasks. Notably, HPNC-DEC seems to perform slightly better than HPNC-IM, but their differences are marginal (less than 0.5) in most cases owing to similar objectives between IM (9) and DEC (13).

## 4.2 Ablation Study

In this section, we further perform ablation studies to confirm the effectiveness of individual design choices of HPNC.

### 4.2.1 Prototype Rotation (RQ2)

When calculating the affinities between nodes and cluster prototypes, we propose to introduce a learnable rotation matrix $R$ applied to these prototypes to allow them to rotate on the hypersphere manifold. To verify whether it's useful to assist the GNN to converge, we replace $R$ in Eq. (6) with an identity matrix such that the coordinates of prototypes are strictly equal to their initial states, *i.e.*, ones randomly initialized and refined by minimizing Eq. (2). The clustering results are reported in Table 2, where the first four rows do not apply rotation and the last four are their counterparts with rotation applied.

We observe that the clustering performances with prototype rotation outperform their counterparts without rotation. Specifically, HPNC-IM and HPNC-DEC have drastic 24.3 and 31.1 NMI drops on the Cora dataset after disabling prototype rotation. What's worse, the NMI on PubMed dataset without rotation is only 9.3 for HPNC-IM and 8.5 for HPNC-DEC, which means the clustering algorithm is not even working. In contrast, they give a plausible performance with rotation enabled, and the standard deviations are much lower. The results strongly suggest that allowing the prototypes to rotate on the hypersphere manifold is indeed helpful and essential. The reason is that we consider pairwise distances of prototypes only in Eq. (2) while their coordinates are ignored. And as a result, unsuitable random initialization of prototypes may fail to match up semantic centroids of node embeddings, leading to performance degradation. Our prototype rotation is thus also regarded as a learning-based strategy to reduce sensitivity to initial states to stabilize training.

### 4.2.2 K-means Labels vs Soft Labels (RQ3)

The clustering results of our proposed HPNC are obtained by directly applying arg max to Eq. (6) so external clustering algorithms are not needed at all. Nonetheless, the learned node embeddings $\hat{Z}$ can be also used for clustering or potentially other downstream tasks. Here, we feed them into K-means and report the clustering results in Table 2 (marked as $\tilde{Z}$+KM). We observe that:

- With prototype rotation applied, the clustering performance of K-means and soft labels are very comparable, which means HPNC can indeed guide the GNN towards discriminative node embeddings so that traditional clustering algorithms can also work well.

- The clustering performance of both K-means and soft-labels degenerate without prototype rotation, but K-means works better than soft-labels in most cases. The reason might be that the autoencoder backbone preserved the clustering structure of the latent embeddings to some extent for K-means to discover, but their geometric centroids deviate far from the predefined prototypes so it's too challenging for HPNC to match them up.

To further investigate the consistency between K-means predictions and soft labels (HPNC-DEC is employed in this experiment), we calculate the proportion of them being the same and plot the results in Figure 2. We

Table 2: Clustering performance of HPNC with different ablation settings. $\tilde{Z}$+KM denotes K-means results of latent embeddings, $Q$ denotes results of Eq. (6).

| Rotate prototype | Scheme | Target | Metric | Cora | CiteSeer | PubMed | ACM | DBLP |
|---|---|---|---|---|---|---|---|---|
| ✗ | HPNC-IM | $\tilde{Z}$+KM | ACC | $60.2_{\pm5.0}$ | $59.9_{\pm10.3}$ | $69.1_{\pm1.0}$ | $80.4_{\pm9.8}$ | $64.4_{\pm4.3}$ |
| | | | NMI | $50.2_{\pm2.7}$ | $37.1_{\pm7.0}$ | $32.3_{\pm1.9}$ | $58.0_{\pm8.3}$ | $35.4_{\pm2.2}$ |
| | | | ARI | $40.0_{\pm3.3}$ | $36.7_{\pm8.4}$ | $31.3_{\pm1.7}$ | $59.9_{\pm11.5}$ | $37.6_{\pm4.3}$ |
| | | $Q$ | ACC | $49.3_{\pm4.0}$ | $44.9_{\pm2.0}$ | $52.5_{\pm6.7}$ | $87.5_{\pm2.5}$ | $63.9_{\pm5.1}$ |
| | | | NMI | $34.6_{\pm7.2}$ | $26.3_{\pm3.3}$ | $9.3_{\pm5.6}$ | $63.0_{\pm5.1}$ | $35.7_{\pm3.0}$ |
| | | | ARI | $26.5_{\pm5.6}$ | $22.4_{\pm3.2}$ | $9.6_{\pm7.3}$ | $67.4_{\pm5.6}$ | $37.8_{\pm5.1}$ |
| | HPNC-DEC | $\tilde{Z}$+KM | ACC | $62.3_{\pm2.9}$ | $61.1_{\pm3.8}$ | $69.3_{\pm1.2}$ | $92.1_{\pm0.2}$ | $64.7_{\pm4.6}$ |
| | | | NMI | $52.5_{\pm1.8}$ | $38.0_{\pm1.4}$ | $32.5_{\pm2.0}$ | $72.1_{\pm0.5}$ | $36.3_{\pm2.5}$ |
| | | | ARI | $42.5_{\pm1.7}$ | $36.6_{\pm2.4}$ | $31.5_{\pm1.9}$ | $78.0_{\pm0.4}$ | $38.8_{\pm4.6}$ |
| | | $Q$ | ACC | $44.9_{\pm2.5}$ | $53.5_{\pm3.5}$ | $52.0_{\pm5.4}$ | $92.1_{\pm0.1}$ | $64.2_{\pm4.5}$ |
| | | | NMI | $28.3_{\pm4.2}$ | $32.4_{\pm3.7}$ | $8.5_{\pm3.7}$ | $72.2_{\pm0.5}$ | $36.1_{\pm2.8}$ |
| | | | ARI | $20.0_{\pm3.2}$ | $27.8_{\pm4.0}$ | $8.3_{\pm5.2}$ | $78.1_{\pm0.4}$ | $38.4_{\pm4.7}$ |
| ✓ | HPNC-IM | $\tilde{Z}$+KM | ACC | $71.4_{\pm2.9}$ | $70.1_{\pm1.5}$ | $69.7_{\pm0.5}$ | $92.1_{\pm0.1}$ | $80.0_{\pm0.4}$ |
| | | | NMI | $57.5_{\pm1.6}$ | $45.4_{\pm0.9}$ | $33.5_{\pm0.6}$ | $72.0_{\pm0.2}$ | $49.8_{\pm0.3}$ |
| | | | ARI | $49.1_{\pm1.4}$ | $47.1_{\pm1.3}$ | $32.3_{\pm0.8}$ | $77.9_{\pm0.2}$ | $55.0_{\pm0.6}$ |
| | | $Q$ | ACC | $74.1_{\pm0.5}$ | $72.4_{\pm0.5}$ | $70.4_{\pm0.4}$ | $92.1_{\pm0.1}$ | $80.0_{\pm0.4}$ |
| | | | NMI | $58.9_{\pm0.6}$ | $46.3_{\pm0.7}$ | $34.6_{\pm0.7}$ | $72.1_{\pm0.3}$ | $49.8_{\pm0.4}$ |
| | | | ARI | $51.5_{\pm0.9}$ | $48.4_{\pm0.7}$ | $33.5_{\pm0.7}$ | $78.1_{\pm0.3}$ | $54.9_{\pm0.6}$ |
| | HPNC-DEC | $\tilde{Z}$+KM | ACC | $70.6_{\pm0.8}$ | $70.6_{\pm2.1}$ | $69.8_{\pm0.5}$ | $92.2_{\pm0.1}$ | $80.2_{\pm0.5}$ |
| | | | NMI | $57.4_{\pm0.5}$ | $46.2_{\pm0.9}$ | $33.8_{\pm0.2}$ | $72.3_{\pm0.3}$ | $50.0_{\pm0.4}$ |
| | | | ARI | $48.8_{\pm1.3}$ | $48.3_{\pm1.5}$ | $32.6_{\pm0.8}$ | $78.3_{\pm0.2}$ | $55.1_{\pm0.7}$ |
| | | $Q$ | ACC | $74.4_{\pm0.5}$ | $73.0_{\pm0.5}$ | $70.6_{\pm0.9}$ | $92.3_{\pm0.1}$ | $80.1_{\pm0.5}$ |
| | | | NMI | $59.4_{\pm0.9}$ | $46.9_{\pm0.7}$ | $33.1_{\pm1.5}$ | $72.4_{\pm0.3}$ | $49.9_{\pm0.5}$ |
| | | | ARI | $51.9_{\pm1.2}$ | $49.7_{\pm0.7}$ | $33.3_{\pm1.4}$ | $78.4_{\pm0.2}$ | $55.0_{\pm0.7}$ |

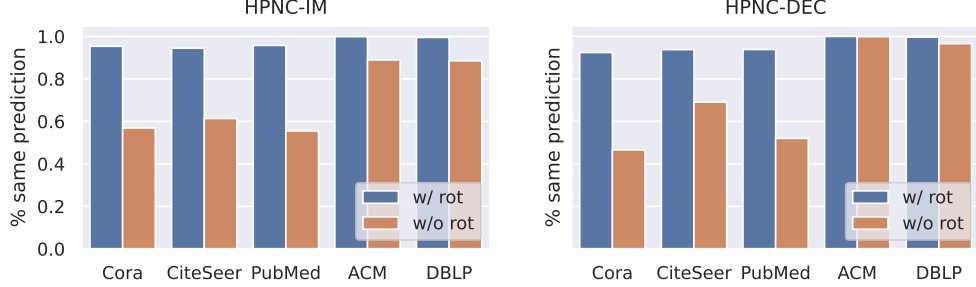

Figure 2: The proportion of Eq. (6) and K-means making the same predictions.

observe that they are highly consistent on all datasets when prototype rotation is applied but deviate far from each other without it. This again confirms our analysis before. To summarize, prototype rotation is a very important component of our proposed HPNC paradigm. It not only helps HPNC to match semantic and predefined prototypes but also leads to more discriminative latent embeddings for potential use as input of other downstream algorithms.

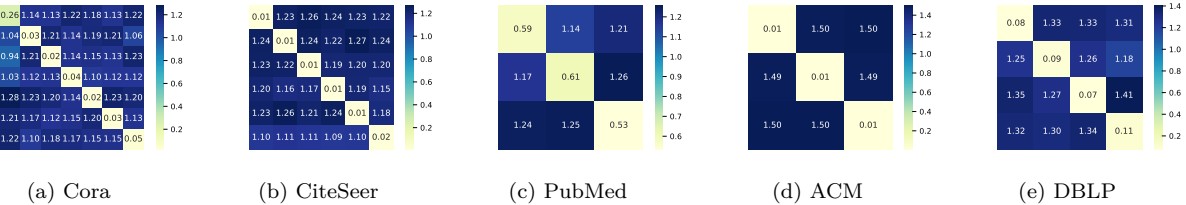

|     |     |     |     |     |
|-----|-----|-----|-----|-----|
| (a) Cora | (b) CiteSeer | (c) PubMed | (d) ACM | (e) DBLP |

Figure 3: Cosine distances between rotated prototypes of HPNC and K-means centroids.

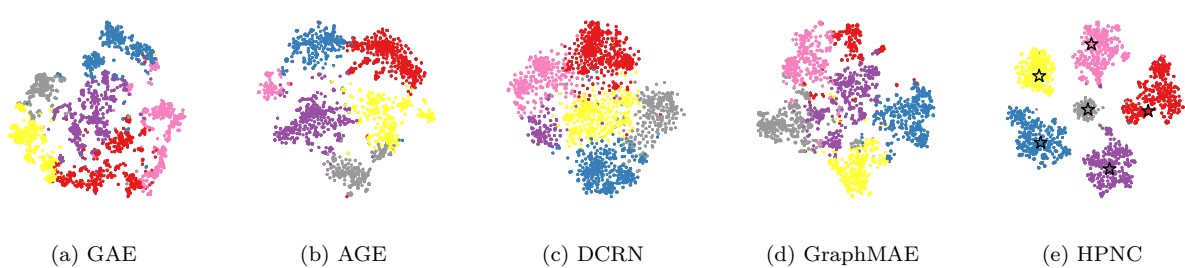

|     |     |     |     |     |
|-----|-----|-----|-----|-----|
| (a) GAE | (b) AGE | (c) DCRN | (d) GraphMAE | (e) HPNC |

Figure 4: t-SNE visualization of learned latent embeddings on CiteSeer dataset. Pentagrams in HPNC denote cluster prototypes. HPNC embeddings are the most discriminative and separable among all the competitors.

### 4.2.3    K-means Centroids vs Prototypes (RQ4)

To investigate the relationships between the clustering centroids inferred from the latent embeddings by K-means and HPNC's (HPNC-DEC is employed in this experiment) rotated prototypes $\{\boldsymbol{R}\tilde{\boldsymbol{\mu}}_j\}_{j=1}^c$, we calculate their pairwise cosine distances and plot the confusion matrices in Figure 3. We observe that K-means centroids and rotated prototypes corresponding to the same cluster distribute very close to each other, which means they are consistent. On the other hand, the pairwise distances between centroids of different clusters are nearly the same, which means different clusters are successfully scattered on the hypersphere manifold. Since K-means centroids are inferred from the latent embeddings, we conclude that HPNC indeed successfully mapped nodes from different clusters to their corresponding prototypes, and made them uniformly distribute on the hypersphere manifold to have large margins.

### 4.3    Visualization (RQ5)

We employ t-SNE (Van der Maaten & Hinton, 2008) to visualize the learned latent embeddings of HPNC (HPNC-DEC is employed in this experiment) on the CiteSeer dataset and compare it with different baseline methods. The results are illustrated in Figure 4. In addition, the cluster prototypes of HPNC are marked in pentagrams. It's obvious that the embeddings learned by HPNC are the most discriminative and separable among all the competitors. Also, data samples from the same cluster distribute evenly around corresponding prototypes, which is a desired property of our motivation.

## 5    Conclusion

This paper presents a novel HPNC paradigm for end-to-end node clustering. In HPNC, uniformly scattered points on a unit-hypersphere are regarded as cluster prototypes. Then, a graph autoencoder is employed to encode the nodes onto the same manifold with the guidance to push them close to the prototypes. Unlike previous works that overlook the separability of learned embeddings, HPNC explicitly maximizes their inter-cluster distances to provide sufficient discriminative power. Moreover, HPNC does not rely on external

clustering algorithms to uncover clustering labels and its objective is fully differentiable, which enables its use as a module in deep learning pipelines. We devise two different schemes based on HPNC to demonstrate its flexibility to work with different clustering backbones. Extensive experiments on real-world graph datasets strongly confirm the effectiveness of HPNC.

**Acknowledgments**

This work was supported in part by the China Postdoctoral Science Foundation under Grant 2022M722532.

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

# A    Appendix: Experimental Setups

## A.1    Datasets

We evaluate the clustering performance on five widely adopted attributed graph datasets: Cora, CiteSeer, PubMed (Yang et al., 2016)[3], ACM and DBLP (Bo et al., 2020)[4].

- Cora is a citation graph of a number of machine-learning papers divided into seven research topics. Each edge represents a citation.

- CiteSeer is a citation graph of scientific publications classified into one of six machine-learning areas. Each edge represents a citation.

- PubMed is a citation graph from the PubMed database, where nodes are papers about three diabetes types and edges are citations among them.

- ACM is a paper graph from the ACM Digital Library. The papers are categorized into three research areas. There is an edge between two papers if they are written by the same author.

- DBLP is an author graph from the DBLP computer science bibliography. Edges denote co-authorship between two authors. The authors are categorized into four research areas.

Statistics of these datasets are summarized in Table 3.

## A.2    Evaluation Metrics

We employ three widely adopted metrics to evaluate node clustering performance, including clustering accuracy (ACC), normalized mutual information (NMI), and adjusted Rand index (ARI). The larger values are, the better the clustering performance.

---

[3]https://github.com/kimiyoung/planetoid
[4]https://github.com/bdy9527/SDCN

Table 3: Statistics for benchmark datasets.

| Dataset | Nodes | Edges | Clusters | Dimension |
|---------|-------|-------|----------|-----------|
| Cora | 2708 | 5278 | 7 | 1433 |
| CiteSeer | 3327 | 4552 | 6 | 3703 |
| PubMed | 19717 | 44324 | 3 | 500 |
| ACM | 3025 | 13128 | 3 | 1870 |
| DBLP | 4057 | 3528 | 4 | 334 |

### A.3   Implementation Details

We implement our proposed HPNC in PyTorch 2.0 and PyG 2.3.0 and use GraphGym (You et al., 2020) for experiment management to ensure reproducibility. Throughout the experiments, the encoders of HPNC are all composed of two GAT layers with 4 128d attention heads and dropout probability 0.1 for attention coefficients. We also apply dropout with probability 0.2 between the GAT layers and use PReLU as the activation function. The decoder is a single GAT layer without non-linear activation. The coefficients $\alpha$, $\beta$ and $\gamma$ in Eqs. (10) and (14) are tuned by random search within the following ranges: $\alpha \in \{0.0, 0.01, 0.02\}$, $\beta, \gamma \in (0.0, 0.1]$. Detailed hyperparameter settings can be found in YAML configuration files from our code.

