# OpenReview forum: "Hyperspherical Prototype Node Clustering"
_TMLR — Accepted by TMLR_

### Review · Reviewer_AWXy · 2023-09-28

**Summary Of Contributions:**

The paper introduces an approach to deep node clustering called Hyperspherical Prototype Node Clustering (HPNC). The proposed method enhances the inter-cluster separability of learned node embeddings by constraining the embedding space to a unit hypersphere. The paper provides a detailed explanation of the HPNC paradigm and its benefits over existing approaches to deep node clustering. It also includes experimental results demonstrating the superior clustering performance of HPNC on various real-world datasets. Overall, the paper presents an interesting technique that has the potential to advance the field of deep node clustering. However, the novelty of the idea is not encouraging. Basically, "enhancing the inter-cluster separability" is a very traditional research point for clustering. It is not clear what is the very unique part of node clustering.

**Audience:**

Yes

**Claims And Evidence:**

Yes

**Requested Changes:**

Cons:
- The major concern is the novelty of the idea is not encouraging. Basically, "enhancing the inter-cluster separability" is a very traditional research point for clustering. It is not clear what is the very unique part of node clustering. The authors are suggested to include more description on this point.
- Some unsupervised node embedding approaches are not cited and compared, such as:
NetWalk: A Flexible Deep Embedding Approach for Anomaly Detection in Dynamic Networks. KDD'18
Learning Deep Network Representations with Adversarially Regularized Autoencoders. KDD'18
It is suggested to have a more thorough literature survey on the graph embedding part.
- The paper does not provide a detailed comparison of the computational efficiency of HPNC compared to other clustering algorithms.
- The paper does not provide a detailed analysis of the robustness of HPNC to noise or outliers in the data.

**Strengths And Weaknesses:**

Pros:
- HPNC enhances the inter-cluster separability of learned node embeddings, which can lead to improved clustering performance.
- The method constrains the embedding space to a unit-hypersphere, which enables the scattering of cluster prototypes over the space with maximized pairwise distances.
- HPNC employs a graph autoencoder to map nodes onto the same hypersphere manifold, which can simplify the clustering process.
- The proposed method is an end-to-end clustering paradigm, which eliminates the need for additional clustering algorithms such as K-means or spectral clustering.
- HPNC has demonstrated competitive clustering performance compared to recent state-of-the-art models on various real-world datasets.

---

> ### Author Response · Authors · 2023-11-19
> **Response to Reviewer AWXy**
>
> Thank you for your recognition of the methodology and its effectiveness. We address your concerns below and aim to provide clarification on some questions.
>
> Q1:
>
> Please find our response in `Explanation for the novelty and core contributions` above.
>
> Q2:
>
> We additionally introduced four notable node embedding methods, including DeepWalk [1], node2vec [2], NetRA [3], and NetWalk [4] in Section 2.1 of the revised manuscript. We also compared with NetRA and NetWalk in clustering experiments.
>
> Q3:
>
> We added Section 3.5 to discuss the computational complexity in the revised manuscript:
>
> When pretraining prototypes, the time complexity of calculating their pairwise
> similarities is $\mathcal{O}(c^2m)$, and there are $cm$ parameters to be updated, so
> the complexity of pretraining is $\mathcal{O}(c^2mt_{pre})$ where $t_{pre}$ refers to
> the number of pretraining epochs. We empirically find that setting $t_{pre}$ to
> 3000 is sufficient, so this step usually finishes in half of a minite.
>
> The complexity of representation learning depends on the specific backbone, which is usually linear with the number of nodes and edges.
>
> The complexity of calculating rotated clustering affinity is $\mathcal{O}(ncm + m^2c)$ by applying rotation to prototypes first and obtain their inner products with samples later. There are $m^2$ trainable parameters in $\boldsymbol{R}$, and its orthogonal parametrization is also $\mathcal{O}(m^2)$. Hence, this step needs $\mathcal{O}((n + m)cmt)$ in total, where $t$ is the number of training epochs.
>
> The complexity of the clustering backbone also depends on the specific choice, which is $\mathcal{O}(nct)$ for both IM and DEC that employed in this work.
>
> In summary, the time complexity of HPNC without representation learning is $\mathcal{O}(c^2mt_{pre} + (n + m)cmt)$. The time complexity of K-means clustering is $\mathcal{O}(ncmt)$, so HPNC is comparable to applying K-means after representation learning.
>
> Q4:
>
> This work focuses on several fundamental issues in deep node clustering. Although the existence of outliers may influence clustering performance, most existing deep clustering methods have no specific response to the outliers [5]. To this end, how to improve the deep clustering robustness to the outliers is still an open research question.
>
> [1] DeepWalk: online learning of social representations. KDD 2014.
>
> [2] node2vec: Scalable Feature Learning for Networks. KDD 2016.
>
> [3] Learning deep network representations with adversarially regularized autoencoders. KDD 2018.
>
> [4] Netwalk: A flexible deep embedding approach for anomaly detection in dynamic networks. KDD 2018.
>
> [5] A comprehensive survey on deep clustering: Taxonomy, challenges, and future directions. arXiv:2206.07579.

---

> ### Comment · Reviewer_AWXy · 2023-11-22
> **All concerns have been addressed**
>
> The authors fully addressed my concerns. It is ready for publication

---

### Review · Reviewer_hnq8 · 2023-10-07

**Summary Of Contributions:**

This work proposes a new method for node clustering tasks. The main innovation of the work is to enforce cluster separation in the optimization objective. Two measures are used to achieve separation:  1) explicitly setting cluster centers and maximizing distances between them; and 2) encouraging the sharpness of the cluster membership matrix. The results indicate the proposed method achieves clear separation and better clustering performance than previous methods.

**Audience:**

Yes

**Broader Impact Concerns:**

No concerns detected.

**Claims And Evidence:**

Yes

**Requested Changes:**

Please check the list of comments in the "Weakness" section.

**Strengths And Weaknesses:**

Strength:

1) The work identifies the problem that clusters need to be separated and then devises optimization terms to encourage separation.

2) The experiment results are generally solid. They verify the hypothesis in the work.


Weakness:

The paper writing should be improved. Here is a list of detailed comments.
1) Section 3.1 should be merged into the Introduction and Related Work sections.
2) Section 3.2.2 is mostly from previous work and not this work's contribution. It should be in a "Background" section
3) Equation (6) it is unclear how R^TR = I is enforced.
4) Section 3.3 introduces new notations y and Q. They should be defined at the problem definition.
5) In figure 1, the word "disambiguate" is not even used in the text. Does it mean "sharpening"?
6) The word "prototype" is not great. It's meaning is "a first, typical or preliminary model of something, especially a machine, from which other forms are developed or copied.", which doesn't seem to be appropriate for a cluster center. Maybe "cluster centers" are better?
7) The experiment settings should be included in the text because they are important for the analysis of the results.

---

> ### Author Response · Authors · 2023-11-19
> **Response to Reviewer hnq8**
>
> Thank you for your recognition of the problem formulation and experiments, and the detailed feedback and suggestions on paper writing. We carefully revised the manuscript and response to the requested changes below.
>
> Q1:
>
> We moved inspirations from metric learning to *Introduction*, and drawbacks of current deep clustering methods to *Related Works*. The rest contents are essentially an overview of the proposed method, so we kept them and renamed the section title to *Overview*.
>
> Q2:
>
> The subsections of Section 3.2 are ordered as how the input data flows through the proposed HPNC framework to help the readers better understand it. In the revised manuscript, we emphasized that Section 3.2.2 is a brief introduction of previous works in its first paragraph to clear up a potential misunderstanding.
>
> Q3:
>
> We initialize $\boldsymbol{R}$ as an identity matrix and enforce the orthogonal constraint with a PyTorch built-in function `torch.nn.utils.parametrizations.orthogonal` during training. The matrix may be parametrized via three different algorithms, where details can be found in PyTorch documentation. We added a footnote to declare this in the revised manuscript.
>
> Q4:
>
> $\boldsymbol{Q}$ was introduced in Eq. (6) right above Section 3.3. $\boldsymbol{y}$ is indeed newly introduced in this section but not used elsewhere, so we added its definition after Eq. (9) of the revised manuscript.
>
> Q5:
>
> Yes, we fixed the figure to keep consistency with the text.
>
> Q6:
>
> Cluster prototype is usually a generalization of cluster centroids. Any representative sample could serve as the prototype of the cluster it belongs. For instance, K-modes clustering [1] uses the modes or most frequent values within each cluster as the prototype. Also, the word "prototype" is widely used in Section 9.1 of [2] when introducing K-means clustering. Nonetheless, we added an explanation in the footnote of the *Introduction* section to avoid possible confusion.
>
> Q7:
>
> We relocated the most critical parts including baselines and evaluation protocol to the beginning to Section 4.
>
> [1] Chaturvedi, Anil, Paul E. Green, and J. Douglas Caroll. "K-modes clustering." Journal of classification 18 (2001): 35-55.
>
> [2] Bishop, Christopher M., and Nasser M. Nasrabadi. Pattern recognition and machine learning. 2006.

---

> > ### Comment · Reviewer_hnq8 · 2023-12-09
> > **Thank you for your responses**
> >
> > I appreciate your effort in addressing my comments. The only point I'd like to mention again is that the method section should not contain much work from the literature. This way of writing helps a reader to identify the contribution of this submission. Section 3.2.2 of the current version still contains a lot of content that is not this submission's innovation. Anyway, I don't insist on this point.

---

### Review · Reviewer_mTj2 · 2023-11-07

**Summary Of Contributions:**

The paper proposes a new node clustering method called Hyperspherical Prototype Node Clustering (HPNC). The general idea is to leverage the clustering-oriented loss with the learned node embeddings to encourage small intra-cluster distances and large inter-cluster distances. Some experiments are conducted to demonstrate the effectiveness of HPNC compared with many baseline methods. Ablation study and analytical results are also provided.

**Audience:**

Yes

**Claims And Evidence:**

No

**Requested Changes:**

In my opinion, the novelty of this work is incremental and the experiments should be improved. Please see the above weakness for any changes.

**Strengths And Weaknesses:**

Strengths
+ It is interesting to improve node embeddings for node clustering by enforcing small intra-cluster distances and large inter-cluster distances. The proposed method looks reasonable to me.

+ Experiments on 5 datasets are conducted to show that HPNC outperforms many baseline methods.

+ The presentation is overall good. The motivation is clearly described and the paper is easy to follow.

I have some concerns regarding the method and experiments.
- The novelty of this paper is incremental to me. Leveraging loss to encourage small intra-cluster distances and large inter-cluster distances is a common idea for improving the clustering algorithm. In addition, it is relatively straightforward that incorporating clustering-based loss into self-supervised loss will improve the clustering performance.

- The improvement of HPNC over baseline methods is not significant. In many cases, the improvements are less than 1%. In addition, the authors should consider comparing more recent graph self-supervised learning baseline methods. Moreover, in the ablation study, it is necessary to report the results of all metrics rather than just NMI. It is also necessary to report the standard deviations of the results over several repeated experiments.

- The current experimental datasets are relatively small. It is better to consider large-scale graph datasets, such as large datasets in OGB.

---

> ### Author Response · Authors · 2023-11-19
> **Response to Reviewer mTj2**
>
> Thank you for your positive appraisal on the problem formulation and presentation. We address your concerns below and aim to provide clarification on some questions.
>
> Q1:
>
> Please find our response in `Explanation for the novelty and core contributions` above.
>
> Q2:
>
> * HPNC is not a specific algorithm with certain network architecture and loss function, but a general framework to incorporate clustering loss and representation learning backbone. Hence, its clustering performance is also related to the chosen backbones.
> * From this perspective, both of the devised schemes HPNC-IM and HPNC-DEC outperform their backbone model GraphMAE, and there's 6.5% improvement of ACC on the DBLP dataset, which is sufficient to reveal the effectiveness of HPNC.
> * Clustering performance is not the only criterion to evaluate a clustering algorithm. Apart from that, HPNC also enjoys additional benefits such as end-to-end training, decoupling from traditional clustering methods, more separable embeddings, etc. These advantages are rigorously verified through ablation studies.
> * Following the suggestions, we additionally compared with a recently published concurrent work [1] and two random-walk-based methods [2][3] (suggested by Reviewer AWXy). We also added ACC and ARI to ablation study, and reported the standard deviations of all experiments in the revised manuscript.
>
> Q3:
>
> * These datasets are widely employed in previous node clustering works including most of the competitors.
> * Handling OGB-scale datasets requires adopting mini-batch training, which is typically non-trivial. Hence, the problem of how to support OGB is essentially how to support mini-batch training.
> * As a general clustering framework, HPNC is naturally applicable to mini-batch training because 1) pretraining and rotating prototypes is irrelevant to the number of samples, and 2) Eq. (6) only involves instance-wise operations and is straightforward to batchify. By incorporating with mini-batch-compatible representation learning and clustering backbones, it could train on OGB-scale datasets, technically.
> * However, the two devised schemes HPNC-IM and HPNC-DEC don't support mini-batch training because their clustering backbones IM and DEC don't. This is the major reason why we didn't evaluate on OGB, and why most of the competitors also didn't (DEC loss is also a building block of them). Developing mini-batch-compatible clustering backbone is beyond the scope of this work. Hence, handling large-scale graph datasets is left for future research.
>
> [1] CONVERT: Contrastive Graph Clustering with Reliable Augmentation. ACMMM 2023.
>
> [2] Learning deep network representations with adversarially regularized autoencoders. KDD 2018.
>
> [3] Netwalk: A flexible deep embedding approach for anomaly detection in dynamic networks. KDD 2018.

---

### Author Response · Authors · 2023-11-19
**Explanation for the novelty and core contributions**

Since both Reviewer mTj2 and Reviewer AWXy raised questions regarding the novelty, we reply to them here to avoid duplication.

* Encouraging small intra-cluster distances and large inter-cluster distances is indeed a general idea for clustering that nearly every clustering algorithm follows. We didn't claim to propose this convention, but focus on **how** to achieve it.
* Roughly speaking, we propose to scatter the cluster prototypes over a unit-hypersphere so their pairwise distances are maximized. Then, we map the data samples onto the same unit-hypersphere and optimize a clustering loss to refine them so that their distributions are expected to agree with the aforementioned convention. The core idea and methodology are elaborated in Sections 3.1 & 3.2. Unlike previous works that employing K-means to obtain initial clustering prototypes *after* self-supervised learning, our prototypes are obtained *before* the learning starts and only allowed to rotate on the unit-hypersphere during training. We are not aware of any previous works adopting similar strategies.
* The core contribution of this work is **how** to incorporate clustering-based loss into self-supervised loss. Sections 3.3 & 3.4 only serves to give two concrete examples based on the proposed framework.

---

### Decision · Action_Editor_y279 · 2024-01-05

**Recommendation:** Accept with minor revision

**Comment:**

Overall, the reviewers agree that this paper is technically correct, and I am confident it is of interest to the TMLR audience. 2 out of 3 reviewers recommended to accept the paper, the 3rd reviewer leaned towards rejection on the grounds of 1) novelty 2) datasets and 3) marginal improvement. After careful consideration I decided to accept the submission for publication, mainly because even though all reviewers noticed to some extent that the novelty may be limited, the TMLR guidelines clearly state that **Papers should be accepted if they meet the criteria [claims/evidence and audience] , even if the contribution or significance of the work is modest. **. I believe that the submission meets both these criteria and hence should published in TMLR.

Please prepare the final version of the manuscript, while doing so, I suggest renaming the new section 3.5 to "Complexity Analysis" (instead of  "Complexity Analyses").

**Audience:**

Yes.

**Claims And Evidence:**

Yes, overall the claims are appropriately supported.